## RESEARCH ARTICLE

# Characterisation of *lmx1b* paralogues in zebrafish reveals divergent roles in skeletal, kidney and muscle development

**Joanna J. Moss[1,‡], Chris R. Neal[2], Erika Kague[1,3], Jon D. Lane[4,*] and Chrissy L. Hammond[1,*]**

## ABSTRACT

LMX1B, a LIM-homeodomain family transcription factor, plays critical roles in the development of multiple tissues, including limbs, eyes, kidneys, brain, and spinal cord. Mutations in the human LMX1B gene cause the rare autosomal-dominant disorder Nail-patella syndrome, which affects development of limbs, eyes, brain, and kidneys. In zebrafish, lmx1b has two paralogues: *lmx1ba* and *lmx1bb.* While lmx1b morpholino data exists, stable mutants were previously lacking. Here, we describe the characterisation of *lmx1b* stable mutant lines, with a focus on development of tissues that are affected in Nail-patella syndrome. We demonstrate that the *lmx1b* paralogues have divergent developmental roles in zebrafish, with *lmx1ba* affecting skeletal and neuronal development, and *lmx1bb* affecting renal development. The double mutant, representing loss of both paralogues (*lmx1b dKO*) showed a stronger phenotype, which included additional defects to trunk muscle patterning, and a failure to fully inflate the notochord leading to a dramatic reduction in body length. Overall, these mutant lines demonstrate the utility of zebrafish for modelling Nail-patella syndrome and describe a previously undescribed role for *lmx1b* in notochord cell inflation.

KEY WORDS: Lmx1b, Zebrafish, Musculoskeletal development, Kidney, Cartilage, Nail-patella syndrome

## INTRODUCTION

LMX1B belongs to the LMX group of the LIM-homeodomain family of transcription factors, a diverse family of regulatory proteins which are characterised by the presence of a homeodomain and two LIM domains (LIM-A and LIM-B) (Curtiss and Heilig, 1998; Hobert and Westphal, 2000). As a transcriptional regulator, LMX1B plays important roles during development in the formation and specification of cell types in multiple tissues, including limbs, eyes, kidneys, brain, ears, and spinal cord (O'Hara et al., 2005; Guo et al., 2007; McMahon et al., 2009; Hilinski et al., 2016; Haro et al., 2017;

[1]School of Physiology, Pharmacology and Neuroscience, University of Bristol, Bristol, BS8 1TD, UK. [2]Wolfson Bioimaging Facility, Faculty of Life Sciences, University Walk, University of Bristol, Bristol BS8 1TD, UK. [3]Centre for Genomic and Experimental Medicine, Institute of Genetics and Cancer, University of Edinburgh, EH4 2XR Scotland, UK. [4]School of Biochemistry, University of Bristol, Bristol, BS8 1TD, UK.
*These authors contributed equally to this work

‡Authors for correspondence ( joanna.moss@bristol.ac.uk; chrissy.hammond@bristol.ac.uk)

J.J.M., 0000-0003-1690-6512; C.R.N., 0000-0001-6604-280X; E.K., 0000-0002-0266-9424; J.D.L., 0000-0002-6828-5888; C.L.H., 0000-0002-4935-6724

Mori et al., 2025), alongside a well-established role in patterning of the dorsal limbs of amniotes (Riddle et al., 1995; Vogel et al., 1995), Cygan et al., (1997). Heterozygous mutations in LMX1B cause Nail-patella syndrome (NPS), a rare genetic disorder which predominantly affects the renal and skeletal systems (Sweeney et al., 2003). In NPS patients, changes to the nails and patellae are defining symptoms, present in 95–100% and 74–95% of patients, respectively (Bongers et al., 2002; Sweeney et al., 2003; Bongers et al., 2005), with renal problems present in 30–50% of affected individuals (Sweeney et al., 2003). However, ocular and neurological changes are also reported in patients (Sweeney et al., 2003).

Given the diverse range of tissues which express *LMX1B,* animal models are needed to provide insights into its tissue-wide expression and the broader impacts of disease-linked mutations. In mice, complete knockout of *Lmx1b* causes postnatal lethality, while heterozygous *Lmx1b* mice show a distinctly different phenotype from human NPS patients who carry similar mutations (Chen et al., 1998). Kidney dysfunction is the most severe symptom in patients with NPS, and usually first presents as proteinuria. In 15% of individuals, this can progress to late-stage kidney disease and renal failure (Sweeney et al., 2003). In mice, *Lmx1b* is highly expressed in kidneys with expression largely limited to the podocytes (Tsuboyama et al., 2016). Podocytes are highly specialised cells which wrap around the outer surface of glomerular capillaries to form a filtration barrier through the interdigitation of their foot processes with neighbouring podocytes (Burghardt et al., 2013). In mice, loss of *Lmx1b* inhibits podocyte development abrogating the formation of podocyte foot processes and the slit diaphragm (Saitoh et al., 2008; Malhotra et al., 2015), with mutant pups born exhibiting severe nephropathies that are fatal within 24 h (Chen et al., 1998; Deegens et al., 2008). In zebrafish, knockdown of *lmx1ba* and *lmx1bb* leads to oedema and altered pronephros patterning in a subset of morphants suggestive of a role for Lmx1b in zebrafish kidney development and function (Burghardt et al., 2013).

Patients with NPS can also develop neuropathy and show increased incidence of epilepsy (Sweeney et al., 2003). Work from animal models has shown that Lmx1b has a role in the developing brain, as it is required for the proliferation, specification, and neuronal differentiation of dopaminergic (DA) and serotonergic (5-HT) progenitors (Ding et al., 2003; Yan et al., 2011), and is essential for the induction and maintenance of the Isthmic Organiser, (Guo et al., 2007; Wever et al., 2019). In both mice and zebrafish, Lmx1b shows high expression in the isthmus and loss of Lmx1b in either model causes abrogation of Isthmic Organiser activity and secretion, leading to abnormal brain morphology and a significant reduction in DA neuronal number in embryos (Adams et al., 2000; Smidt et al., 2000; O'Hara et al., 2005; Guo et al., 2007).

Along with changes to patellae and nails, other skeletal changes form a core part of the NPS phenotype. These include a loss of flexion

around distal joints in the limbs, which is thought to be caused by the absence or hypoplasia of key muscles and ligaments at these joint sites due to defective limb patterning (Sweeney et al., 2003; Lovelace and May, 2021). From early animal studies, a key role for *Lmx1b* in limb patterning and development has already been well-established. For example, in chicks, overexpression of *Lmx1b* causes ventral to dorsal transformation of limb mesoderm (Vogel et al., 1995), while loss of *Lmx1b* in mice leads to a loss dorsal patterning in limbs resulting in symmetrical ventral–ventral patterning of footpads, muscles, tendons and ligaments, and loss of dorsal hair follicles (Chen et al., 1998). However, recent genome-wide association studies using data from the UK Biobank have identified polymorphisms in *LMX1B* that are associated with osteoarthritis, a finding that has since been replicated in Chinese populations (Chen et al., 2024). As a disease typically associated with ageing, these results indicate a role for LMX1B beyond developmental stages, that may encompass other tissues within the musculoskeletal system.

Although not considered a standard part of the NPS phenotype, abnormal development of the proximal musculature has been observed, with some NPS patients showing decreased muscle mass in the upper arms and legs (Sweeney et al., 2003). Mouse *Lmx1b*$^{-/-}$ models also show limb muscle abnormalities, especially in the paws, where patterning of the muscles and ligaments is disrupted; although it is likely that these are secondary to skeletal patterning defects (Cross et al., 2014). Alternatively, these could relate to changes to motor neuron innervation to the limbs which is observed when *Lmx1b* expression is altered (Kania et al., 2000). However, these phenotypes have yet to be described in detail, and the functional impact of these patterning defects have not been investigated.

Whilst stable *lmx1b* knockout zebrafish lines have been previously developed to study ear and neural phenotypes, a full characterisation of loss of *lmx1b* in zebrafish is yet to be reported (Schibler and Malicki, 2007; Obholzer et al., 2012; Hilinski et al., 2016). In this study, we have further explored the developmental roles of *lmx1b* using a zebrafish vertebrate model. Due to a genome-wide duplication event thought to have occurred before the divergence of zebrafish, pufferfish, and medaka lineages (Amores et al., 1998; Gates et al., 1999; O'Hara et al., 2005), zebrafish express two paralogues of *lmx1b*: *lmx1ba* and *lmx1bb*. Most studies in zebrafish have used morpholinos (short RNA constructs injected directly into the yolk of embryos at the single-cell stage) to temporarily knock down the expression of these two paralogues of *lmx1b* during early development (O'Hara et al., 2005; Filippi et al., 2007; McMahon et al., 2009; Obholzer et al., 2012; Burghardt et al., 2013). These models have been useful for demonstrating the expression pattern and role of both *lmx1b* paralogues in zebrafish which appear very similar to mammalian models (McMahon et al., 2009). But given the neonatal lethality of mouse *Lmx1b* knockout models and the lack of a full *lmx1b* knockout zebrafish model, we looked to generate and stable *lmx1ba lmx1bb* knockout zebrafish lines and to characterise phenotypes in non-neural and aural tissues.

Using these lines, we show that the paralogues of *lmx1b* have largely divergent roles in zebrafish development, with *lmx1ba* deletion affecting skeletal development, and *lmx1bb* affecting kidney development. Loss of both paralogues resulted in muscular abnormalities and limited body growth, concomitant with defects in vacuolated cell inflation in the notochord. Together, these results demonstrate the utility of these lines to study the role of *lmx1b* in developmental processes and highlight a new potential function for *lmx1b* in notochord development and trunk muscle formation in zebrafish.

## RESULTS

### Establishment and characterisation of *lmx1b* single and double mutants

Using CRISPR-Cas9, we generated two *lmx1b* knockout lines targeting the two *lmx1b* paralogues in zebrafish (Fig. 1A), which were crossed to generate a double *lmx1b* knockout (termed *dKO*). The *lmx1ba*$^{-/-}$ fish have a 7 bp deletion in the highly conserved homeobox domain, leading to a 27 amino acid truncation to the Lmx1ba protein (246 amino acids in wild type, to 219 in *lmx1ba*$^{-/-}$) (Fig. S1A-B, *left*). Meanwhile, the *lmx1bb*$^{-/-}$ fish have a 19 bp insertion within the conserved LIM A domain, predicted to cause a 335 amino acid truncation of the Lmx1bb protein (375 amino acids in wild type, to 40 in *lmx1bb*$^{-/-}$) (Fig. S1A-B, right). No major morphological changes were detected in *lmx1ba*$^{-/-}$ larvae (Fig. 1B), but *lmx1bb*$^{-/-}$ and *dKO* larvae showed high incidence of oedema, which worsened by 7 dpf (83%/72% with incidence of oedema at 5 dpf in *lmx1bb*$^{-/-}$ and *dKO*, respectively; Fig. 1B, black arrowheads, and C), as well as a lack of swim bladder inflation by 5 dpf (Fig. 1B, white asterisks; 89%/100% no swim bladder inflation in *lmx1bb*$^{-/-}$ and *dKO*, respectively, Fig. 1D). All *lmx1b* mutants also showed no changes to otolith number by 5 dpf, although the morphology was altered in 41–52% of otoliths in *lmx1bb* and *dKO* mutants, respectively (Table S1). Strikingly, from 2 dpf, the *dKOs* developed a truncated body phenotype (Fig. 1B, white arrowhead) and slower trunk growth (Fig. 1E). Given these severe phenotypes, both the *lmx1bb*$^{-/-}$ fish and *dKOs* did not survive past 13 dpf and 10 dpf, respectively. The *lmx1ba*$^{-/-}$ fish were viable and survived to adulthood (Fig. 1F). Our results demonstrate that *lmx1bb*, but not *lmx1ba*, is essential for zebrafish survival. The differences in morphology between the *lmx1b* mutant lines suggest a degree of functional divergence and genetic compensation between the paralogues, such that the double mutants have trunk phenotypes not observed in either single mutant line.

### Loss of *lmx1bb* disrupts kidney formation

As the *lmx1bb*$^{-/-}$ larvae showed a high incidence of oedema at 5 dpf, this phenotype was followed to 7 dpf and analysed further by electron microscopy (Fig. 2). In studies performed in mice and from human NPS patients, *Lmx1b* has been strongly associated with kidney development and formation (Chen et al., 1998; Burghardt et al., 2013), with mutations in *LMX1B* causing kidney dysfunction in most NPS patients (Bongers et al., 2002). In zebrafish, oedema can be a sign of defective kidney function as the kidneys play a key role in excreting water and maintaining proper osmoregulation (Hill et al., 2004; Outtandy et al., 2019). By 5 dpf, the majority of *lmx1bb*$^{-/-}$ larvae (84%) had mild to moderate oedema which worsened in severity by 7 dpf (Fig. 2A). Oedema of the yolk sac was most prominent (Fig. S2, black arrowhead), followed by pericardial oedema and oedema of the eye by 4–5 dpf (Fig. S2, white arrowhead, and pink arrowhead, respectively). Therefore, ultrastructural analysis was used to explore the structure of the renal system in *lmx1bb*$^{-/-}$ and wild-type larvae (Fig. 2B-E).

In zebrafish, glomerular filtration begins by 2 dpf (Drummond and Davidson, 2010), and by 4 dpf, all common features of the kidney glomerulus, such as fenestrated endothelial cells, a glomerular basement membrane, and podocytes, can be observed via transmission electron microscopy, suggesting establishment of the filtration barrier (Kramer-Zucker et al., 2005) (Fig. 2B). In the wild-type glomerulus, key structures forming the filtration barrier are easily identified and show a clear organisation at 6 dpf (Fig. 2C). Surrounding blood vessels and endothelial cells, a dark, electron dense line can be seen indicating formation of the glomerular

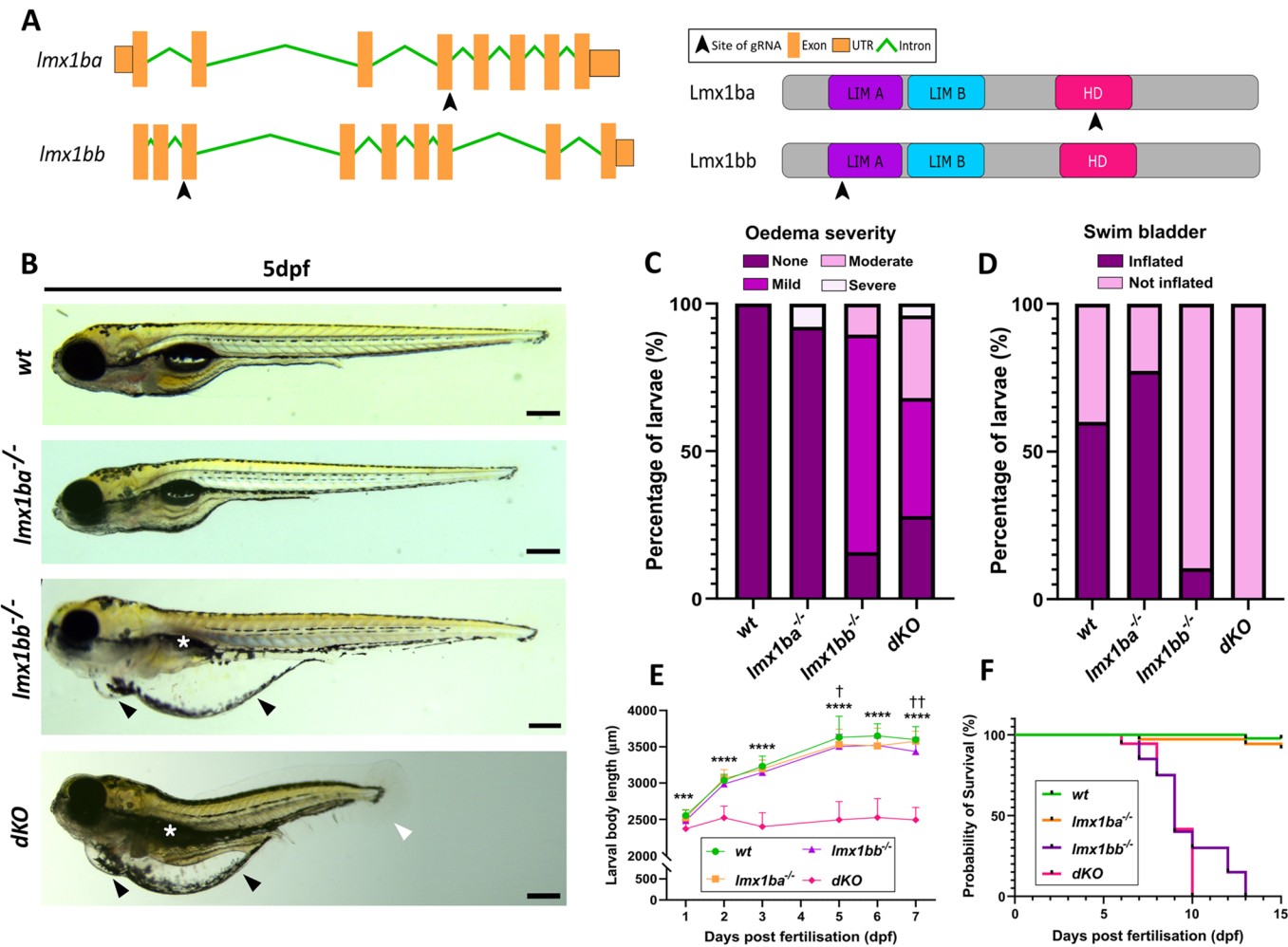

**Fig. 1. Two _lmx1b_ mutant fish lines were generated using CRISPR-Cas9.** (A) Schematic showing exons targeted by gRNAs (black arrowheads) in gene and protein sequences for _lmx1ba_ and _lmx1bb_. HD, homeobox domain. (B) Stereomicroscope images of wild type and _lmx1b_ mutant lines at 5 dpf. Black arrowheads indicate oedema; white asterisks show loss of swim bladder inflation in _lmx1bb_$^{-/-}$ and _dKO_; white arrowhead shows body truncation in _dKO_. Scale bars: 250 µm. Quantification of number of larvae at 5 dpf showing (C), oedema at varying severities and (D) swim bladder inflation or non-inflation. _N_=19 for wild type; 38 for _lmx1ba_$^{-/-}$; 19 for _lmx1bb_$^{-/-}$ and 25 for _dKO_. (E) Graph showing body length measurements from 1–7 dpf in wild-type, _lmx1ba_$^{-/-}$, _lmx1bb_$^{-/-}$ and _dKO_ larvae. Two-way ANOVA performed comparing to wild type, where *** $P$=0.009 and ****$P$<0.0001 for _dKO_; † $P$=0.0458 and †† $P$=0.0064 for _lmx1bb_$^{-/-}$. (F) Kaplan–Meier survival curve showing survival rate of _lmx1b_ mutant lines compared to wild type _N_=14 for _wt_ and _lmx1ba_ mutants; 20 for _lmx1bb_ mutants; 36 for _dKO_ mutants.

basement membrane (GBM) (Fig. 2Ci, white dashed lines). On the urinary side of the GBM, podocyte foot processes appear as tall columnar-like structures along the edge of the GBM (Fig. 2C, white arrowheads), with interdigitation between foot processes present in some areas (Fig. 2Cii, IDP, pink and white arrowheads). On the blood side of the GBM, fenestrations in the endothelium, which allows for the filtration of low-molecular-weight waste products from circulation, can be seen next to the GBM (Fig. 2Cii, pink asterisks).

By contrast, in the _lmx1bb_$^{-/-}$ larvae, the cells and structures forming the glomerulus have a disrupted organisation, with some structures being entirely absent (no GBM, Fig. 2D) or incomplete (loss of proper podocyte interdigitation, white asterisk, Fig. 2D). Here, podocyte foot processes are apparent (_FP_, Fig. 2D), but few are interdigitating with other foot processes (white arrowheads, Fig. 2D), resulting in a disrupted filtration barrier. The mitochondria appeared round and swollen (_M_, Fig. 2D), indicative of mitochondrial dysfunction and a sign of kidney disease (Qi et al., 2017). In other areas, there are fragments of a GBM along an endothelial boundary (black arrowheads, gbm, Fig. 2E) between the

urinary space and blood vessel, suggesting partial formation of a GBM. However, podocyte foot processes and endothelial fenestrations along this boundary line are harder to identify and appear more swollen and disordered (Fig. 2E, purple arrowheads). Meanwhile, some areas show complete loss of glomerular structure and patterning (Fig. S3) indicating that there is a spectrum of disruption level to kidney development which mirrors the range of oedemic severity seen by 7 dpf. These changes indicate that loss of _lmx1bb_ disrupts kidney structure, and GBM and podocyte formation, which is likely decreasing kidney function and causing oedema in the _lmx1bb_$^{-/-}$ larvae. Overall, these suggest a divergence in gene function whereby _lmx1bb_, but not _lmx1ba_, is required for kidney development in zebrafish.

### Loss of _lmx1ba_ delays cartilage growth and chondrocyte maturation

Given the skeletal changes observed in NPS patients (Beals and Eckhardt, 1969; Vogel et al., 1996), the effect of loss of _lmx1ba_ and _lmx1bb_ on skeletal formation in zebrafish was explored. Larvae

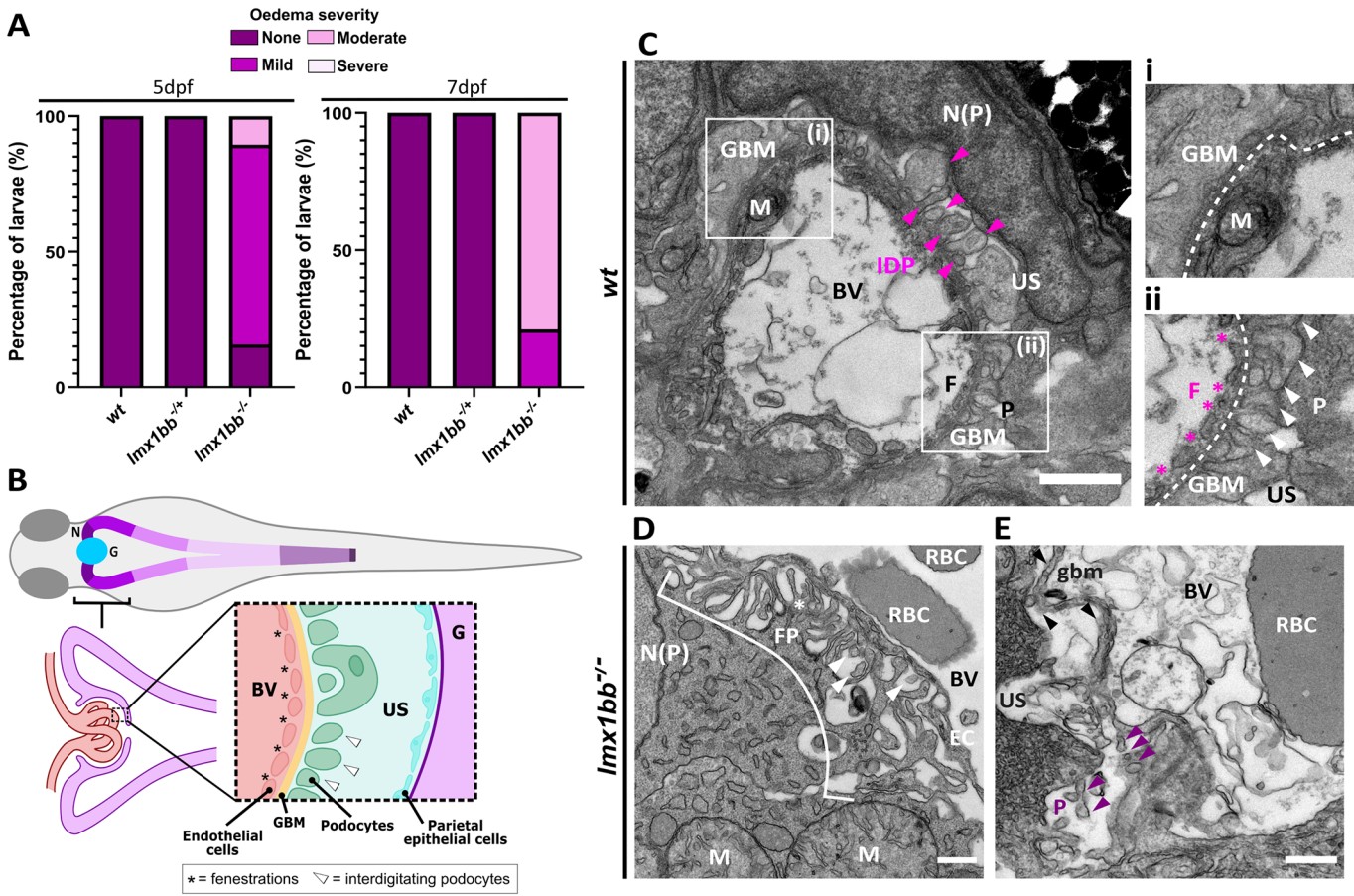

**Fig. 2. Loss of *lmx1bb* disrupts kidney formation and function.** (A) Graphs showing percentage of *lmx1bb* larvae within each oedema classification at 5 dpf (left) and 7 dpf (right). *N*=9 for wild type; 16 for *lmx1bb*⁻/⁺; 19 for *lmx1bb*⁻/⁻ mutants. (B) Schematic showing normal layout of kidney glomerulus. Electron micrographs images of the glomeruli of (C) wild type, and (D,E) *lmx1bb* mutants at 6 dpf. For C, pink and white arrowheads indicate interdigitating podocyte foot processes, white dashed lines show GBM, and pink asterisks indicate fenestrations. For D and E, white and purple arrowheads indicate possible interdigitating foot processes, white bracket shows region of foot processes. BV, blood vessel; EC, endothelial cell; F, fenestrations; FP, foot processes; G, glomerulus; GBM, glomerular basement membrane; gbm, partial glomerular basement membrane; IDP, interdigitating podocyte foot processes (indicated by alternating foot processes of different electron densities); M, mitochondria; N, nephron; N(EC), nucleus of endothelial cell; N(P), nucleus of podocyte cell; P, podocytes; RBC, red blood cell; US, urinary space. Scale bars: 1 µm for C; 500 nm for D and E. *N*=3 for both.

were immunostained with Col2a1 and the lower jaws from each mutant were measured at 3 dpf and 5 dpf (Fig. 3A-D; *lmx1bb* data not shown). While the *lmx1bb*⁻/⁻ larvae showed no changes to jaw shape compared to wild type, the lower jaws of the *lmx1ba*⁻/⁻ larvae were significantly shorter, consistently showing decreased lower jaw length and width from 3 dpf to 5 dpf (Fig. 3B-D).

To explore the cause behind the decrease in cartilage growth, an EdU assay was performed (Fig. 3E,F). Cells positive for both EdU and Col2a1 were counted across the whole jaw and at the lower jaw joint (Fig. 3G,H). At both locations, the *lmx1ba*⁻/⁻ larvae showed a reduction in chondrocyte proliferation compared to wild type. To ascertain whether loss of *lmx1ba* was affecting chondrocyte maturation as well as proliferation, ultrastructural analysis of wild type and *lmx1ba*⁻/⁻ chondrocytes was performed (Fig. 3I-N). At 5 dpf, the wild type chondrocytes appeared elongated with sparsely filled cytoplasm (Fig. 3I-J), which is an indicator of late-stage hypertrophy in chondrocytes. Along the length of the cartilage, the majority of wild type chondrocytes were neatly stacked in a column, giving the cartilage a smooth edge border (Fig. 3I). In comparison, the *lmx1ba*⁻/⁻ chondrocytes were more electron dense, and had a significantly greater number of vesicles per chondrocyte compared to wild type (Fig. 3K-L). The *lmx1ba*⁻/⁻ chondrocytes also showed disrupted organisation, with more chondrocytes not fully

intercalating (Fig. 3K, pink arrowheads). These data show that by 5 dpf, *lmx1ba*⁻/⁻ chondrocytes are less hypertrophic than wild type, suggesting a delay in chondrocyte maturation in *lmx1ba* mutants. Therefore, overall, loss of *lmx1ba* leads to reduced chondrocyte proliferation and maturation, which together impact cartilage growth.

### Lmx1b is required for notochord cell inflation
Lmx1b *dKO* larvae showed a severe truncation in trunk length from 3 dpf. To test what might cause this, larvae were immunostained for the skeletal muscle marker A4.1025 at 3 dpf and 5 dpf, (Fig. 4A-E). The *lmx1ba*⁻/⁻ and *lmx1bb*⁻/⁻ larvae showed comparable trunk muscle formation to wild type. However, the *dKOs* showed a range of affected muscle phenotypes (Fig. 4C-E), with most larvae showing a moderate to severe muscle defect (Fig. 4F). The *dKOs* showed a high incidence of gaps between myofibrils (Fig. 4C,D, pink asterisks), and branching of myofibrils perpendicular to the organisation of the somite fibres (Fig. 4D, white arrowheads). And in rare cases, complete loss of slow muscle organisation, with fibres forming no discernible pattern or structure within somites (Fig. 4E).

To analyse these effects on trunk muscle formation, changes to somite size and fibre organisation were quantified (Fig. 4G,H). The *dKOs* showed significantly smaller somites and decreased muscle coverage (quantified as the green fluorescence percentage coverage

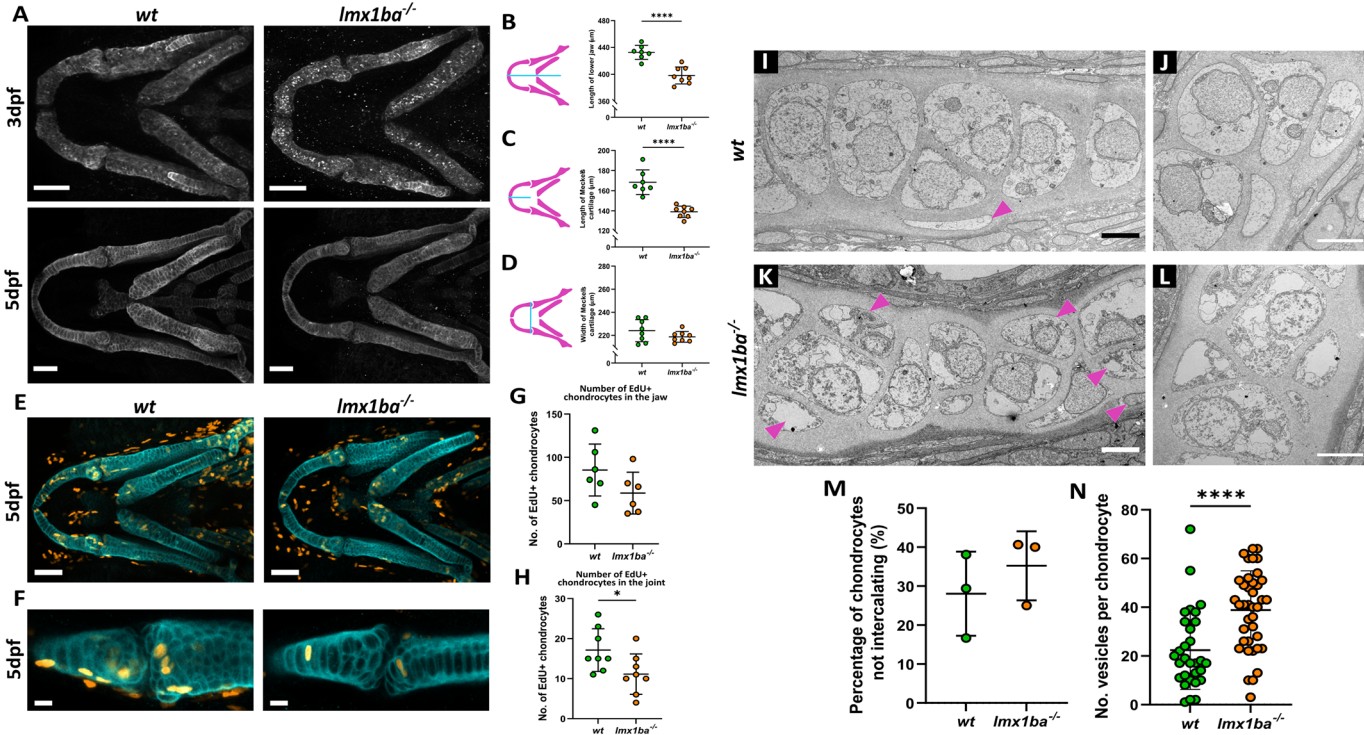

**Fig. 3. Loss of *lmx1ba* delays cartilage growth and maturation.** (A) Confocal images of z-stack projections of the lower jaw at 3 dpf and 5 dpf in wild type and *lmx1ba* mutants immunostained for Col2a1 (white). Scale bars: 50 µm. Schematics indicating where measurements were made for (B) length of lower jaw, (C) length of Meckel's cartilage and (D) width of Meckel's cartilage (*left*), with quantification of results (right). *N*=7 for wild type; 8 for *lmx1ba* mutants. Student's unpaired *t*-test performed for all where ****$P$<0.0001. Confocal z-stack projections of the (E) lower jaw and (F) lower jaw joint at 5 dpf in wild type and *lmx1ba* mutants, immunostained for EdU (cyan) and Collagen Type II (Col2a1; red) following 24-h treatment with EdU Click-iT. Scale bars: 50 µm for E and 10 µm for F. Number of EdU-Col2a1-positive cells quantified for (G) the whole lower jaw and (H) jaw joint. *N*=6 for wild type; 8 for *lmx1ba* mutants. Student's unpaired *t*-test performed for all where *$P$=0.0362. (I-L) Electron microscopy of ethmoid plate in wild type and *lmx1ba* mutants at 5 dpf. (I,K) Pink arrowheads highlight areas of non-uniformity and non-intercalating chondrocytes. Scale bars: 5 µm. (M) Number of chondrocytes on periphery of cartilage that are not aligning down central line of stack. Calculated as percentage of total cell number along ethmoid plate in one section per fish. (N) Number of vesicles present per chondrocyte. Mann–Whitney unpaired *t*-test performed where, ****$P$=0.0001. *N*=30 chondrocytes total from three larvae per genotype.

per somite) compared to wild type, indicating reduced fibre alignment and greater gaps between fibres. As somite number in the *dKOs* is comparable to wild type, these data suggest that the reduction in body length may be due to reduced myofibril elongation in the trunk. Investigation of the craniofacial musculature by myosin and thrombospondin-4 immunostaining showed that loss of *lmx1b* in double mutants has no obvious effect on jaw musculature (fibre number, position or length) or ligament development or formation (Fig. 4I-J). This further corroborates the idea indicates that the muscle phenotype observed in *dKOs* is limited to the trunk muscle. To investigate whether other factors are affecting body elongation, we next looked at the notochord.

In embryonic zebrafish, the notochord is formed of two cell types: the outer notochord sheath cells, which secrete an extracellular matrix (ECM) to surround the notochord, and inner vacuolated notochord cells, formed of individual, fluid-filled vacuoles (Ellis et al., 2013b). During early development, these vacuoles inflate rapidly and expand to fill the notochord sheath (Ellis et al., 2013a). It has been shown in zebrafish that the morphogenic force of the inflated vacuoles is required for body axis elongation, and that loss or damage to these cells can cause body shortening and later kinking in the spine (Ellis et al., 2013a). Average notochord width was measured for individual wild type and dKO larvae at 3 dpf and 5 dpf (Fig. 5A-B). At both 3 dpf and 5 dpf the dKOs showed a significant decrease in average notochord width, suggesting that the vacuolated cells within the notochord are not inflating normally. Using the lipid dye

BODIPY, the membrane edges of the tightly packed vacuolated cells could be clearly seen in the wild type notochord (Fig. 5C). In the dKOs, the vacuolated cells were smaller, less tightly packed and more difficult to distinguish (Fig. 5C). To gain clarity, low magnification electron microscopy images of a cross section through the notochord in 3 dpf wild-type and dKO larvae were taken (Fig. 5D-E). In wild type, fluid filled vacuolated cells with clear membrane boundaries can be seen (pink cells in inset), with smaller, more electron-dense cells lining the inner side of the notochord sheath (green cells in inset, Fig. 5Di). In the *dKOs*, the vacuolated cells were less inflated and still contained some organelles and vacuoles (highlighted purple in inset, Fig. 5Ei). Uninflated, electron dense cells could also be seen within the centre of the notochord (orange in inset, Fig. 5Ei). The cells lining the sheath (false coloured green in inset, Fig. 5Ei), were also larger and less well organised, while the notochord sheath (highlighted in yellow, Fig. 5Ei) appeared thicker. These data indicate that loss of *lmx1b* has a negative effect on vacuole cell formation and could be impeding vacuole inflation, leading to reduction in notochord extension and overall body growth. This in turn may be altering muscle formation and elongation, although the disrupted muscle fibre patterning in some *dKOs* suggests a separate role for *lmx1b* in muscle formation which will need to be investigated further.

## DISCUSSION

In this study, we generated and characterised two new *lmx1ba* and *lmx1bb* knockout lines in zebrafish. We show that *lmx1ba* and

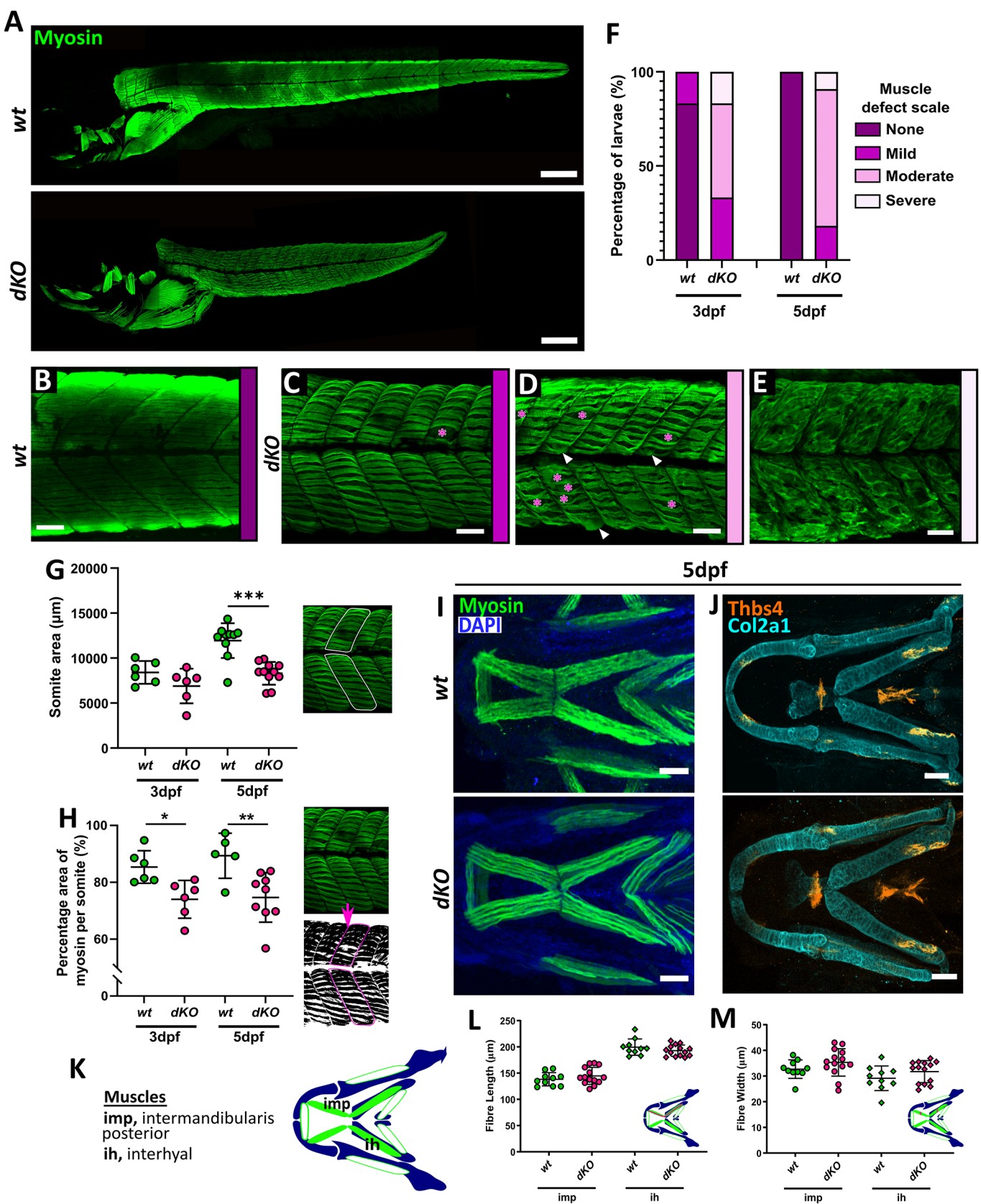

**Fig. 4.** See next page for legend.

*lmx1bb* have divergent roles in zebrafish development, affecting skeletal development (*lmx1ba*), and kidney development (*lmx1bb*), respectively. Further, loss of both paralogues resulted in reduced body growth, concomitant with defects in vacuolated cell inflation in the notochord. These phenotypes, although divergent, align with the phenotype of *Lmx1b* mutant mouse models and NPS patient

**Fig. 4. Loss of Lmx1b affects muscle formation in the trunk but not in the head.** Confocal z-stack projections of *wt* and *dKO* mutants at 5 dpf immunostained with myosin, showing the whole body, scale bars: 250 μm, (A) and zoomed in regions of trunk muscle where defects in muscle formation increase in severity from no defect in wild type (B) to increasing number of defects in *dKOs* (C-E). Scale bars: 50 μm. (C,D) Pink asterisks indicate large gaps between myofibrils; (D) white arrowheads indicate abnormal fibre branching. (F) Percentage of larvae showing muscle defects at different severities quantified for *wt* and *dKO* larvae at 5 dpf. Differences in muscle formation between *wt* and *dKO* larvae at 3 dpf and 5 dpf quantified as (G) somite area and (H) the percentage coverage of muscle (myosin) staining per somite across all genotypes. Each data point is an average of measurements taken from three separate somites per fish. Schematics (right) show how measurements were made. *N*=11 for wild type and *dKO* mutants. Welch's *t*-test performed for G where ***P*=.0001. Student's unpaired *t*-tests performed for H where **P*=.01 and ***P*=.0092. (I) Confocal maximum projections of *wt* and *dKO* mutant larvae at 5 dpf immunostained for muscle (green) and co-stained with DAPI (blue). Scale bars: 100 μM. (J) Confocal z-stack projections of the lower jaw of wild type and *dKO* mutants immunostained for the tendon marker Thbs4 (cyan) and Collagen type II (Col2a1; red) at 5 dpf. Scale bars: 50 μM. (K) Schematic showing location and names of key muscles in the lower jaw of zebrafish. Muscles measured here are filled in in green. (L) Graphs showing quantification of muscle fibre length and width in the intermandibularis posterior (imp) and interhyal (ih) muscles at 5 dpf. *N*=5 for wild type and 7 for *dKOs*. Two datapoints plotted per fish, per age as measurements taken from right and left side of the lower jaw. Student's unpaired *t*-test performed at each age for all graphs.

symptoms, indicating that these lines are relevant for the study of *lmx1b*. Although previous zebrafish studies have indicated differences between *lmx1ba* and *lmx1bb* expression during development (McMahon et al., 2009; Burzynski et al., 2013; He et al., 2014; Hilinski et al., 2016), here functional divergence

between the paralogues is demonstrated and characterised across the whole body for the first time.

Although there can be several causes of oedema in zebrafish such as defects to the lymphatic (Karpanen and Schulte-Merker, 2011), cardiac (Miura and Yelon, 2011) or renal systems (Outtandy et al., 2019), the strong association of *Lmx1b* with kidney development and maintenance suggest that defective kidney function is the most likely cause here. The prevalence of oedema in the *lmx1bb*[−/−] larvae but not in the *lmx1ba*[−/−] larvae correlates with morpholino studies where the single knock-down of *lmx1bb* resulted in severe oedema, whilst the single knock-down of *lmx1ba* gave a less penetrant phenotype (Burghardt et al., 2013). The high incidence of oedema in the *dKOs* supports this finding, as loss of both paralogues resulted in a more severe phenotype.

In the *lmx1bb*[−/−], the incidence of oedema was striking as it was found to occur in all larvae by 7 dpf, including larvae showing no observable developmental defects at 5 dpf. This indicates that loss of *lmx1bb* has a severe effect on kidney formation and that this phenotype is completely penetrant. This was confirmed by ultrastructural analysis which showed a disorganised and disrupted glomerulus morphology in the *lmx1bb*[−/−] larvae by 6 dpf. GBM formation was largely disrupted or when present, showed irregular thickening, while the number of podocyte foot processes appeared reduced, with some areas showing a 'moth-eaten' appearance similar to that seen in *Lmx1b* knockout mouse models and NPS patients (Dreyer et al., 1998; Kolhe et al., 2002). In *lmx1bb*[−/−] larvae, mitochondria also appeared round and swollen, a sign of mitochondrial dysfunction, which is a recognised sign of kidney pathology in human disease (Qi et al., 2017). This loss of structure would have a severe effect on kidney function and the proper regulation of water within the body. Therefore,

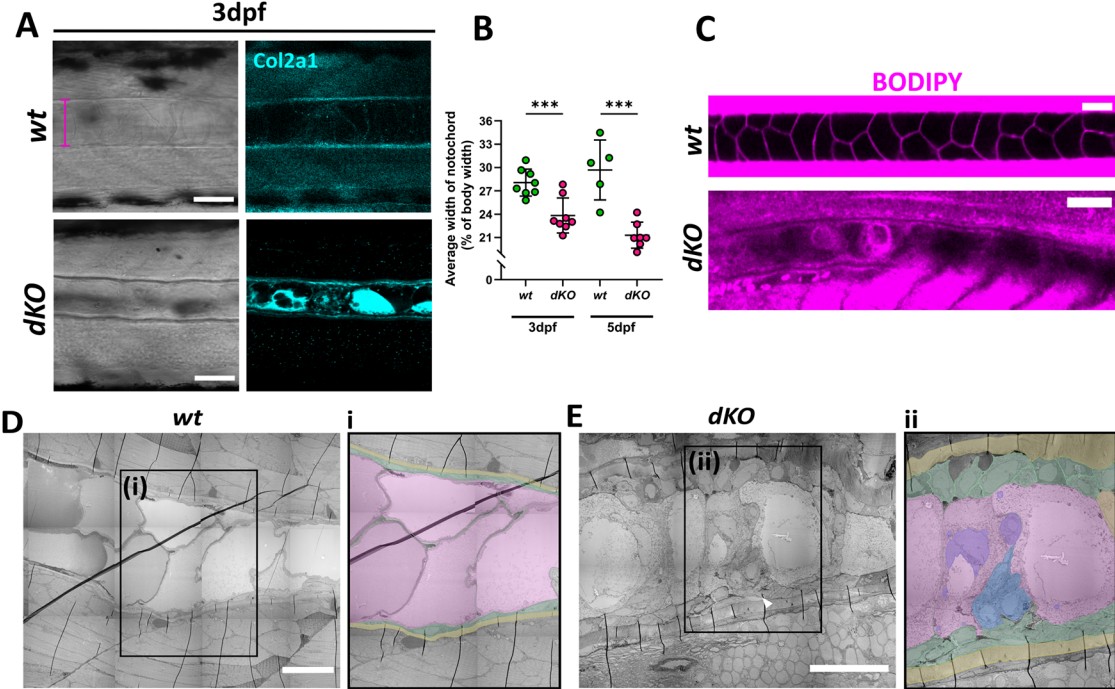

**Fig. 5. Lmx1b is required for notochord cell inflation and body elongation.** (A) Confocal images of the notochord in wild type and *dKO* larvae immunostained for Col2a1 at 3 dpf to show notochord sheath and aid measurement of notochord width. Pink line shows where notochord width was measured from. Scale bar: 50 μm. (B) Graph showing width of notochord as a percentage of body width at 3 dpf and 5 dpf. *N*=7 for wild type at 3 dpf and 5 at 5 dpf; 8 for *dKO* at 3 dpf and 7 at 5 dpf. Student's unpaired *t*-test performed at each age, where *** *P*=0.0009 and *** *P*=0.0004, respectively. (C) Confocal images of BODIPY stained larvae at 3 dpf. Scale bar: 50 μm. Low power electron microscopy images of notochord in (D) wild type and (E) *dKO* larvae at 3 dpf. Scale bar: 25 μm. Insets (Di, Eii) show false colouring for vacuolated cells (pink); unvacuolated cells (blue); nearby sheath cells (green); notochord sheath (yellow) and intracellular contents of vacuolated cells (purple).

Biology Open

this demonstrates an essential role for *lmx1bb* in kidney development, which is required for survival in zebrafish.

In the developing skeleton we identified a role for *lmx1ba* but not *lmx1bb* as having a role in skeletal development. The *lmx1ba*$^{-/-}$ larvae showed reduced chondrocyte proliferation, organisation and maturation to hypertrophy, which likely underpins the reduced growth of the jaw as hypertrophication of chondrocytes is one of the main drivers required for skeletal growth (Chen and Long, 2014). Recently, Mori et al., showed that during ear canal development, *lmx1ba* and *lmx1bb* are positive transcriptional regulators of two key ECM components: hyaluronic acid (HA) and Versican (Mori et al., 2025). By interacting together, HA and Versican can modulate the biophysical properties of ECM, indicating that Lmx1b expression could be important for ECM synthesis and formation during ear development, which could extend to other ECM-rich components such as jaw cartilage or the notochord sheath (Mori et al., 2025).

The *dKOs* showed a striking reduction in trunk growth beyond 2 dpf despite normal somite number. Analysis of trunk muscle revealed changes to the organisation of slow muscle fibres, which was not seen in the craniofacial musculature. This is not unexpected, as craniofacial muscle and trunk muscle develop from different cell progenitor sites in the developing embryo; craniofacial muscle being derived from neural crest cells, and trunk muscle from the paraxial mesoderm which differentiates to form the somites (Kaucka et al., 2016; Liu et al., 2022).

At both 3 dpf and 5 dpf, myofibrils in the *dKOs* were shorter, as expected, but also displayed large gaps between fibres and branching of myofibrils perpendicular to the normal direction of growth, with the phenotype worsening with age. Strikingly, neither single *lmx1ba*$^{-/-}$ or *lmx1bb*$^{-/-}$ larvae gave any indication of trunk muscle abnormalities. This indicates either a high degree of compensation between the paralogues, or that this muscle phenotype is secondary symptom to another growth abnormality that is triggered only by the loss of both *lmx1b* paralogues. The inflation and expansion of vacuolated cells within the notochord is known to be essential for body elongation during development (Ellis et al., 2013a). The decreased notochord width in the *dKOs* and observed differences in vacuolated cell size and organisation compared to wild type indicate that this process is not occurring which would account for the lack of body growth beyond 2 dpf. Additionally, the presence of smaller and rounder vacuoles in *dKO* notochords is reminiscent of the phenotype seen in zebrafish with mutations in genes responsible for vacuolated cell function, such as H$^{+}$-ATPase (mutant line name, *atp6v1e1b*$^{hi577aTg}$) and the sodium-dependent neutral amino acid transporter, Slc38a8b, which regulates vacuolated cell volume (Ellis, 2014). Therefore, together these indicate that a vacuolated cell defect is present in the *dKOs*, however, the mechanism behind this defect still needs to be identified.

In addition to the key developmental roles for Lmx1b illustrated in diverse animal models, conditional double *Lmx1b/Lmx1a* knockout in the mouse brain has revealed key roles for Lmx1b in the maintenance of the populations of neurons that express these genes (Laguna et al., 2015). Crucially, this study highlighted potential roles for Lmx1b as a transcription factor for several key autophagy and lysosomal genes, with evidence of an autophagic flux phenotype in degenerating neurons (Laguna et al., 2015). Lmx1b was subsequently shown to also control the expression of nuclear-encoded mitochondrial genes (Doucet-Beaupré et al., 2016). Interestingly, a direct, molecular link to the autophagy pathway was later revealed in which LMX1B binds to members of the ATG8 family of autophagy regulators through a conserved LC3-

interacting region (or LIR), with ATG8 s acting as co-factors for LMX1B (Jiménez-Moreno et al., 2023). Such reciprocal cooperativity with the autophagy system should therefore be considered when assessing developmental and/or maintenance roles of LMX1B in any model. It is thus noteworthy that our study has revealed clear organelle retention phenotypes in chondrocytes and in notochord cells in *dKO* larvae (Fig. 3K-L; Fig. 5Ei). Indeed, in the developing jaw, *dKO* larvae had phenotypes that were clearly reminiscent of an autophagy-deficient *atg13*$^{-/-}$ zebrafish model we recently described (Moss et al., 2021), providing further evidence of possible interplay between *Lmx1b* and the autophagy pathway. Taking these observations together, these results demonstrate that the *lmx1ba*$^{-/-}$ and *lmx1bb*$^{-/-}$ zebrafish lines closely follow the expected phenotype of an *LMX1B* knockout model. With the survival of all mutant lines into late larval stages, and the fact that zebrafish develop externally, these lines enable the effect of loss of *lmx1b* on body formation to be observed all the way through development and into the formation of a functional animal. As demonstrated here, the functional effects of these *lmx1b* mutations on key systems and organs can also be examined, unlike other rodent models. Therefore, the *lmx1ba*$^{-/-}$, *lmx1bb*$^{-/-}$ and *dKO* lines generated and characterised here provide an alternative and representative model for the study of NPS and the role of *lmx1b* in developmental processes.

## MATERIALS AND METHODS
### Zebrafish husbandry and CRISPR-mutant lines
Zebrafish (*Danio rerio*) were raised and maintained under standard conditions (Aleström et al., 2020). Experiments were approved by the local ethics committee (the Animal Welfare and Ethical Review Body of the University of Bristol) and performed under UK Home Office project licence.

Stable *lmx1ba* (*lmx1ba*$^{bsl2962}$) and *lmx1bb* (*lmx1bb*$^{bsl2816}$) zebrafish lines were generated using CRISPR-Cas9 mutagenesis, as we have previously performed for other genes (López-Cuevas et al., 2021). Briefly, gRNAs targeting exon 4 of *lmx1ba* (ENSDART00000162334.2) and exon 3 of *lmx1bb* (ENSDART00000076420.3) were synthesised (Merck): CRISPR-O7 *lmx1ba* – CGGUUUGGACGAGACCUCGAAGG; CRISPR-O1 *lmx1bb* – GAUGGCACUCCGUCCCUGAAGGG, CRISPR-O2 *lmx1bb* - GACGGAGUGCCAUCACCAGGCGG, CRISPR-O3 *lmx1bb* – AGCG-GUCGGAUAUCGGCCGCUGG. *lmx1ba* or *lmx1bb* gRNAs (20 pmol/µl) were incubated with SygRNA Tracr RNA (20 pmol/µl; Sigma, TRACRRNA05N) and GeneArt Platinum Cas9 nuclease (600 ng/µl; Invitrogen), followed by injections into zebrafish embryos at one-cell stage. Three gRNAs O1-3 were injected simultaneously to target *lmx1bb*.

The efficiency of CRISPR mutagenesis was determined using fragment analysis from DNA extracted from single F0 embryos (Carrington et al., 2015). F0s were raised to 3 months and crossed to wild-type fish (TL/EKK strain) to generate heterozygous F1s. Sanger sequencing of PCR products confirmed adult F1 heterozygous carriers, which showed a 7-base pair (bp) deletion at 635-641 bp (212aa, Chr5:5032833, GRCz11/danRer11) in *lmx1ba*, and a 19-base pair insertion in *lmx1bb* at 85 bp (31aa, Chr8:33258580, GRCz11/danRer11). Both cause a premature stop codon, which leads to a truncation at amino acid 219 for *lmx1ba* and 40 for *lmx1bb*. Heterozygous *lmx1ba*$^{-/+}$ and *lmx1bb*$^{-/+}$ fish were in-crossed to generate stable homozygous (*lmx1ba*$^{-/-}$ or *lmx1bb*$^{-/-}$) mutants that were used in experiments.

### DNA extraction and genotyping
For DNA extraction whole embryos or caudal fin tissue (Wilkinson et al., 2013) was incubated in base solution (25 mM NaOH, 0.2 mM EDTA) for 30 min at 95°C before addition of equal volume of neutralisation solution (40 mM Tris–HCl, pH 5.0). For genotyping, DNA of *lmx1ba* and *lmx1bb* were amplified using the following primers: *lmx1ba* Forward (F): ATGTGAAGCCGGAGAAAGG, *lmx1ba* Reverse (R): GGTGGC-CAGCTTGTATGACT; *lmx1bb* set 1 F: CGTGCGCATTACACCAATAA, *lmx1bb* set 1 R:TGTAAAACGACGGCCAGTCGTAATATTATTCGGA-CGCCTTT; *lmx1bb* set 2 F: TTACCCTTCAGGGACGGAGT, *lmx1bb* set 2

R: GCCAAATTCTTGGACAAACG. Amplicons were digested using BsaI for *lmx1ba* (R3733, New England Biolabs, NEB) and DraIII for *lmx1bb* primer set 1 (R3510, NEB), respectively, or for *lmx1bb* primer set 2, the amplicons were sent for Sanger sequencing. The digestion of *lmx1ba* amplicon resulted in in 130 bp and 140 bp (seen as one band, wild-type allele) and 275 bp (mutant allele). For *lmx1bb,* the digestion results in 400 bp (wild-type allele) and 225 bp and 275 bp (mutant allele).

### Larval survival assays
For survival assays, larvae were genotyped and separated into tanks based on genotype. Larvae were monitored twice daily and dead larvae counted and removed until 15 dpf.

### Cell proliferation assay
Cell proliferation was measured using the Click-iT EdU imaging kit (Invitrogen) according to the manufacturer's instructions. Briefly, 5 dpf larvae were treated with 400 µM EdU in Danieau's buffer for 24 h. Larvae were fixed in 4% PFA overnight at 4°C, washed, and then incubated in the Click-iT reaction cocktail for 30 min, following the manufacturer's instructions.

### Whole-mount immunohistochemistry
Larvae at 1–5 dpf were fixed in 4% PFA then stored at −20°C in 100% MeOH. Larvae were re-hydrated, washed in PBS-T (PBS with 0.1% Triton X-100) before permeabilisation with 10–15 µg/ml proteinase K (4333793, Sigma) at 37°C. For A4.1025 antibody staining, larvae were instead permeabilised in 0.25% Trypsin for 15 min on ice. Next, samples were blocked in 5% horse serum and incubated with primary antibodies overnight at 4°C [anti-myosin (clone A4.1025) (sc-53088, 1:500, Insight Biotech, UK); anti-col2a1 (II-II6B3-s, 1:20, DSHB, IA, USA); anti-thrombospondin-4 (Thsb4) (ab211143, 1:500, Abcam, UK)]. Samples were washed extensively in 1× PBS-T before incubation with Alexa-Fluor secondary antibodies (Invitrogen) diluted at 1:500 in 5% horse serum. Incubations with secondary antibody were performed for 2 h at room temperature, in the dark. For DAPI staining, larvae incubated in PBS-T containing DAPI (1 µg/ml) for 1 h at room temperature before washing.

### BODIPY live staining
BODIPY-TR methyl ester (C34556, Molecular Probes, USA) was used for live imaging of the notochord. Briefly, larvae at 3–5 dpf were placed in 25 µM BODIPY-TR in Danieau's buffer for 1 h, rinsed three times for 5 min then imaged.

### Confocal imaging of larvae
Larvae were mounted in 1% LMP agarose (16520050, Thermofisher, MA, USA) and imaged using a Leica SP5-II AOBS tandem scanner confocal microscope attached to a Leica DMI 6000 inverted epifluorescence microscope and oil immersion 20× or 40× objectives run using Leica LAS AF software (Leica, Germany). Maximum projection images were assembled using LAS AF Lite software (Leica) and Fiji (Schindelin et al., 2012).

### Stereomicroscope imaging of zebrafish
Images of live larvae from 1–7 dpf were obtained using a Leica MZ10 F modular stereo microscope system at 1–8.3× magnification. For live imaging, fish were anaesthetised using 0.1 mg/ml MS222 (Tricaine methane sulfonate) diluted in Danieaus and imaged laterally.

### Transmission electron microscopy (TEM)
Larvae at 3–6 dpf were fixed in 2.5% glutaraldehyde in 0.1 M sodium cacodylate buffer (pH 7.3) at 4°C, washed and then fixed in 0.2 M osmium in sodium cacodylate buffer with 1.5% potassium ferrocyanide for 1 h at room temperature. After washing, larvae were placed into tissue processor for transmission electron microscopy (TEM; Leica EMTP, Leica Microsytems UK Ltd) using a standard Epon resin protocol. Briefly, larvae were stained with uranyl acetate and Walton's lead aspartate solution then dehydrated in ethanol and infiltrated with Epon resin via propylene oxide and polymerised at 60°C for 48 h.

Ultra-thin sections (70 nm) of Epon embedded larvae were cut using a diamond knife, collected on Pioloform coated one-hole copper grids (AGS162, Agar Scientific, UK) and observed using a Tecnai 12-FEI 120 kV BioTwin Spirit transmission electron microscope. Images of chondrocytes were taken from transverse sections of the ethmoid plate, whilst kidney images were taken from lateral sections (*n*=3 larvae per genotype). Images taken using an FEI Eagle 4k×4k CCD camera and analysed using the freehand selection tool and multi-point counter in Fiji (Schindelin et al., 2012).

### Image analysis and statistics
Standard larval lengths were obtained from lateral images at 1–7 dpf by measuring from the nose of the fish to the end of the tail manually in Fiji. Jaw and muscle measurements were made manually in Fiji using the straight line tool from max projections of confocal z-stacks. For somite area and percentage muscle coverage per somite, measurements from three individual somites were taken and an average calculated. For somite area, somites were drawn around using the polygon tool and for percentage muscle coverage, a threshold between 10–45 (depending on image brightness) was applied to the drawn around area and the value for percentage area taken. Location of where measurements were taken from is shown in the corresponding figures.

For EdU cell proliferation counts, EdU-positive chondrocytes positive for Col2a1:mCherry were counted by going through z-stack. For whole jaw measurements, muscle fibre number and length were measured manually using Fiji and maximum projections of confocal z-stacks of *wt* and *lmx1b*$^{-/-}$ larvae immunostained with anti-Col2a1 or A4.1025 antibodies, respectively.

Statistical analyses were performed using Graphpad Prism v.9. Error bars on all graphs represent the mean±standard deviation.

### Acknowledgements
We would like to acknowledge Elizabeth Blyth and Jan Stanka, for their help with the larval imaging during their undergraduate projects. We thank the staff of the Wolfson Bioimaging Facility for imaging support, particularly Dr Stephen Cross for his help with image analysis and in the development of a modular analysis program. We also thank Mathew Green and technical staff from the University of Bristol's Animal Scientific Unit (ASU) for providing zebrafish husbandry.

### Competing interests
The authors declare no competing or financial interests.

### Author contributions
Conceptualization: J.J.M., C.L.H., J.D.L., E.K.; Data curation: J.J.M.; Formal analysis: J.J.M., C.R.N.; Funding acquisition: C.L.H., J.D.L.; Investigation: J.J.M., E.K., C.R.N.; Methodology: J.J.M., C.L.H., J.D.L., E.K.; Project administration: C.L.H., J.D.L.; Supervision: C.L.H., J.D.L.; Validation: J.J.M.; Visualization: J.J.M.; Writing – original draft: J.J.M., C.L.H., J.D.L.; Writing – review & editing: J.J.M., C.L.H., J.D.L., E.K.

### Funding
J.J.M. was funded by the Wellcome Trust Dynamic Molecular Cell Biology PhD Programme at the University of Bristol (083474/Z/07/Z). J.J.M., J.D.L. and C.L.H. received funding from the BBSRC (BB/Y002504/1). C.L.H. and E.K. were funded by Versus Arthritis Senior Fellowship (21937). E.K. was funded by Versus Arthritis Career Development Fellowship (23115). C.R.N. was funded by the University of Bristol Wolfson Bioimaging Facility. Open Access funding provided by the UKRI and Wellcome Trust. Deposited in PMC for immediate release.

### Data and resource availability
Data are available at the University of Bristol data repository, data.bris, at https://doi.org/10.5523/bris.37pf1srsatzy22k456lebbe5tf.

### Peer review history
The peer review history is available online at https://journals.biologists.com/bio/article-lookup/doi/10.1242/bio.062038.reviewer-comments.pdf

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
