## [Peer Review File · Biology Open]

Characterisation of *lmx1b* paralogues in zebrafish reveals divergent roles in skeletal, kidney and muscle development

Chrissy L. Hammond, Jon D. Lane, Erika Kague, Chris R. Neal and Joanna J. Moss
DOI: 10.1242/bio.062038

Editor: Tristan Rodriguez

Review timeline

Original submission:	28 April 2025
Editorial decision:	2 May 2025
First revision received:	23 June 2025
Accepted:	30 June 2025

Original submission

First decision letter

MS ID#: bio.062038

MS Title: Characterisation of *lmx1b* paralogues in zebrafish reveals divergent roles in skeletal, kidney and muscle development

Authors: Joanna J Moss; Chrissy L Hammond; Jon D Lane; Erika Kague; Chris R Neal

Dear Dr Moss,

I have now reached a decision on the above manuscript.

The reviewer reports are shown at the bottom of this email or can be accessed, together with a copy of this decision letter, by going to:

As you will see, the reviewers gave favourable reports, but raised some critical points that will require amendments to your manuscript. I hope that you will be able to carry these out, because we would like to be able to accept your paper.

Reviewer 1

Comments for the author

This is a nicely written and illustrated manuscript describing two new zebrafish lines harbouring mutations in *lmx1ba* and *lmx1bb*. The Introduction is very comprehensive and the authors have discussed previous findings not just in zebrafish but also in mouse mutants and human patients harbouring mutations in *LMX1B*. The M&M is written in a very clear manner and with sufficient detail to reproduce the experiments. The Results section is clear, informative and the Figures are of high quality and properly labelled. The Discussion is informative and not too long. The description of these two new zebrafish lines is relevant, have provided new phenotypes and will be helpful for future studies. Overall, I think this is very nice piece of work.

I have some minor comments/suggestions:

1. I may have missed it but for the non-specialised it could be helpful to add an asterisk in the swim bladder.

2. Line 254. The functional divergence is clear. The genetic compensation is inferred. I just wonder whether the authors have any data showing the expression of the *lmx1bb* is unregulated in *lmx1ba*^{-/-} mutants or vice versa?? This is not required for publication, but if the authors have available data, it could complement nicely the statement.
3. Line 313. The size of the chondrocytes in the *lmx1ba*^{-/-} mutants is smaller. This is clear in the Picture too (Figure 3K-L). However, in Figure 3 M,N, rather than assessing the size, the authors have quantified the intercalation and vesicles, but not the size of the chondrocytes. Is there a reason for this? If possible, a quantification of the size could be shown as well, to match the statement in the text (line 313).
4. Line 337. Figure 6I-J, rather than Figure 4I-G.
5. Line 337. The following statement: "This indicates that the muscle phenotype observed in dKOs is limited to the trunk muscle, and that changes to trunk musculature may not be causal for the reduced body length but may be a secondary effect". Could the authors explain the reasons why? If the head muscles and the trunk muscles have a different origin (i.e. neural crest and paraxial mesoderm), why the defects in the trunk must be secondary? Also, if the *lmx1ba*^{-/-} mutants have smaller jaws, are the jaw muscle smaller? In line 336, it is stated that the dKO has no defect in the jaw musculature, but are they shorter due to the under-development of the jaw cartilage?
6. In the text, eg line 340, the *col2a1* staining is not presented, but it is shown in Figure 5A. What has this staining revealed?
7. Suppl Fig 1. It could help to state that the deletion or insertion of a few nucleotides either destroys or creates restrictions sites, which can be used to genotype the fish as shown in C.

Reviewer 2

Comments for the author

Moss et al., describe the phenotype of both the individual zebrafish *lmx1ba* and *bb* mutant, and using a morpholine, the combined *lmx1b* mutant. The study does well in determine the redundant and non-redundant phenotypes of the mutant. I would have liked to have seen some attention paid to the otocyst (a site of *lmx1b* function) and my only suggestion is that a comment be included on the number of otoliths seen in the mutants. More discussion of the results from Munjal's group (<https://doi.org/10.1242/dev.203003>) should be made, and the authors should see how to incorporate those finding in this study.

Experimental quality

Does each figure have the proper controls?

- The manuscript describes mutant phenotypes. The comparison with wild type controls is justified.

Are experiments performed using appropriate methods that will answer the question (or test the hypothesis or support the observations) posed by the authors? Is the right tool used for the job?

- Experiments are appropriate and in general the conclusions drawn are fine. For describing the phenotypes in the podocytes, the cartilage and in the notochord, the authors use TEM studies. These are really well-done, It might be worth supplementing this with marker studies, For suggesting that there is a delay in the formation of hypertrophic cartilage, could this be assessed using a marker (CollX, Indian hh). Similarly, the notochord phenotype appears to be a result of delayed/incomplete maturation - are antibody markers available?

Were the data analyzed using appropriate statistical tests?

- Appropriate statistical tests were performed.

Reproducibility

Were experiments in each figure performed using adequate number of biological replicates?

- Yes. N's are stated throughout the manuscript.

Is there sufficient raw data to assess the rigor of the analysis?

- Yes.

Does the methods section provide sufficient detail to permit reproducibility?

- Yes.

Completeness

Are the author's conclusions supported by the data?

- The conclusions drawn are supported by the data, with the caveat stated above.

Are there any flaws in the experimental design that invalidate the approach taken by the authors?

-The study is well done.

Are there experiments that have not been performed, but if true would disprove the conclusion? If yes, and if such experiments would be costly or time-consuming to perform, do the authors acknowledge this in a discussion of the limitations?

- No. Although if a line on the otocyst phenotype could be added, that would support the usefulness of the MO approach in analysing

Scholarship

Do the authors cite and discuss the merits of relevant data that would argue against their conclusion?

- Yes

Do the authors cite and discuss the merits of relevant data that would support their conclusion?

- It would be useful to discuss the data from Munjal's group (<https://doi.org/10.1242/dev.203003>) on versican control by Lmx1b. Could this be a general role (in regulating ECM components) for Lmx1b types? The study does show, well, the distinction in the expression of the two subtypes and suggests a function that seems similar to what the authors are describing here.

For techniques/methods manuscripts, Do the authors cite and discuss the current state of the field and clearly explain how the method improves the field?

- N/A

Reviewer's Responses to Questions

Experimental quality

Does each figure have the proper controls?

If 'No', please indicate reasons in Comments for Author box below.

Reviewer #1:

Yes

Reviewer #2:

Yes

Were the data analyzed using appropriate statistical tests?

If 'No', please indicate reasons in Comments for Author box below.

Reviewer #1:

Yes

Reviewer #2:

Yes

Reproducibility

Were experiments performed using adequate number of biological replicates?

If 'No', please indicate reasons in Comments for Author box below.

Reviewer #1:

Yes

Reviewer #2:

Yes

Does the methods section provide sufficient detail to permit reproducibility?

If 'No', please indicate reasons in Comments for Author box below.

Reviewer #1:

Yes

Reviewer #2:

Yes

Completeness

Are the manuscript's conclusions supported by the data?

If 'No', please indicate reasons in Comments for Author box below.

Reviewer #1:

No

Reviewer #2:

Yes

Scholarship

Do the authors cite and discuss the merits of data that would argue for and against their conclusion?

If 'No', please indicate reasons in Comments for Author box below.

Reviewer #1:

Yes

Reviewer #2:

Yes

Does the manuscript title & abstract accurately reflect the contents of the manuscript, without hyperbole?

If 'No', please indicate reasons in Comments for Author box below.

Reviewer #1:

Yes

Reviewer #2:

Yes

First revisionAuthor response to reviewers' comments

The authors would like to thank both reviewers for their thorough reading of the manuscript, their positive comments and helpful suggestions. We have pasted the comments which required amendments below, along with our responses. Any changes to the manuscript text have been highlighted in yellow.

Reviewer 1's comments

1. I may have missed it but for the non-specialised it could be helpful to add an asterisk in the swim bladder. **We thank Reviewer 1 for highlighting this omission, we have added in an asterisk to show the location of the swim bladder in Figure 1B.**
2. Line 254. The functional divergence is clear. The genetic compensation is inferred. I just wonder whether the authors have any data showing the expression of the *lmx1bb* is unregulated in *lmx1ba*^{-/-} mutants or vice versa?? This is not required for publication, but if the authors have available data, it could complement nicely the statement. **We thank Reviewer 1 for this suggestion; while we agree this would be beneficial information we don't have these data.**
3. Line 313. The size of the chondrocytes in the *lmx1ba*^{-/-} mutants is smaller. This is clear in the Picture too (Figure 3K-L). However, in Figure 3 M,N, rather than assessing the size, the authors have quantified the intercalation and vesicles, but not the size of the chondrocytes. Is there a reason for this? If possible, a quantification of the size could be shown as well, to match the statement in the text (line 313). **We have quantified chondrocyte size and found that although the *lmx1ba*^{-/-} chondrocytes are smaller compared to wt, the difference is not significant (see figure 1 below). Therefore, we have amended line 320 to reflect this.**

Figure 1 - Area of chondrocyte cytoplasm calculated by subtracting nuclear area from total chondrocyte area.

4. Line 337. Figure 6I-J, rather than Figure 4I-G. **Figure number has been updated.**
5. Line 337. The following statement: "This indicates that the muscle phenotype observed in dKO is limited to the trunk muscle, and that changes to trunk musculature may not be causal for the reduced body length but may be a secondary effect". Could the authors explain the reasons why? If the head muscles and the trunk muscles have a different origin (i.e. neural crest and paraxial mesoderm), why the defects in the trunk must be secondary? **Given the changes to notochord cell inflation in the dKO, and that being a key driver for body elongation, it makes sense that the muscle phenotype is a secondary effect of this. We hypothesise that the muscle fibres appear 'loose' because they are not being stretched as much as normal due to loss of body growth. However, this could be made clearer in the text, and we have removed the statement in line 337 comparing trunk and craniofacial muscle.**
6. Also, if the *lmx1ba*^{-/-} mutants have smaller jaws, are the jaw muscle smaller? In line 336, it is stated that the dKO has no defect in the jaw musculature, but are they shorter due to the under-development of the jaw cartilage? **We have not measured the jaw musculature of the *lmx1ba*^{-/-} mutants as there was no indication that the musculature of the fish was affected as the trunk musculature is normal. In the dKO, the jaw musculature is comparable length and width to *wt*. We have added this data to Figure 4 to make this more clear (Figure 4K-M).**
7. In the text, eg line 340, the *col2a1* staining is not presented, but it is shown in Figure 5A. What has this staining revealed? **The *col2a1* staining was used to show more accurately where the notochord sheath was for measurements. A line to this effect has been added to the figure legend to explain this better, line 731. We think the difference in *col2a1* appearance between the *wt* and dKO larvae is due to lack of notochord cell inflation making the notochord sheath more deflated/less turgid. So, instead of seeing a clean cut through the notochord sheath (like cutting through a pipe, as in the *wt*), in the dKO cross-section we are seeing part of the outer sheath on the lateral side, as well as the dorsal and ventral edges. This is just our hypothesis and is unconfirmed which is why this comment is not included in the manuscript.**
8. Suppl Fig 1. It could help to state that the deletion or insertion of a few nucleotides either destroys or creates restriction sites, which can be used to genotype the fish as shown in C. **Suppl fig 1 figure legend has been updated to include this extra information.**

Reviewer 2's comments

1. I would have liked to have seen some attention paid to the otocyst (a site of *lmx1b* function) and my only suggestion is that a comment be included on the number of otoliths seen in the mutants. **We thank Reviewer 2 for this suggestion; we have counted the number of otoliths in *wt* and *lmx1b* mutants and they all have 2 otoliths, although a proportion of the otoliths in the *lmx1bb* and dKO mutants appear deformed (please see table below). These data have been added to a table in the Supplemental data and a comment to this effect has been added to the manuscript (line 255).**

Table 1 - Otolith number is unaffected, but morphology is altered in *lmx1b* mutants compared to *wt* at 5dpf.

	Total fish	Otolith no.			Normal?		Normal as a %	
		2	1	0	Y	N	Y	N
wt	19	19	0	0	19	0	100%	0%
lmx1ba ^{-/-}	28	28	0	0	27	1	96%	4%
lmx1bb ^{-/-}	17	17	0	0	10	7	59%	41%
dKO	23	23	0	0	11	12	48%	52%

2. More discussion of the results from Munjal's group (<https://doi.org/10.1242/dev.203003>) should be made, and the authors should see how to incorporate those findings in this study.

And, it would be useful to discuss the data from Munjal's group on versican control by Lmx1b—could this be a general role (in regulating ECM components) for Lmx1b types? The study does show, well, the distinction in the expression of the two subtypes and suggests a function that seems similar to what the authors are describing here. **A discussion of data from Mori et al., has been included in the discussion (line 410).**

3. For describing the phenotypes in the podocytes, the cartilage and in the notochord, the authors use TEM studies. These are really well done, it might be worth supplementing this with marker studies. For suggesting that there is a delay in the formation of hypertrophic cartilage, could this be assessed using a marker (CollX, Indian hh). Similarly, the notochord phenotype appears to be a result of delayed/incomplete maturation - are antibody markers available? **We thank Reviewer 2 for this suggestion and agree that it would add to the paper. Unfortunately, after trialling three colX antibodies, we found that none of them worked for immunostaining on the jaw or notochord.**

Second decision letter

MS ID#: bio.062038R1

MS Title: Characterisation of lmx1b paralogues in zebrafish reveals divergent roles in skeletal, kidney and muscle development

Authors: Joanna J Moss; Chrissy L Hammond; Jon D Lane; Erika Kague; Chris R Neal

Dear Dr Moss,

I am happy to tell you that your manuscript has been accepted for publication in Biology Open, pending our standard publication integrity checks. It was accepted on 30 Jun 2025.